



# High-resolution hydrometeorological and snow data for the Dischma catchment in Switzerland

Jan Magnusson[1], Yves Bühler[1], Louis Quéno[1], Bertrand Cluzet[1], Giulia Mazzotti[1], Clare Webster[1,2], Rebecca Mott[1], Tobias Jonas[1]

[1]WSL Institute for Snow and Avalanche Research SLF, Davos, Switzerland
[2]Department of Geosciences, University of Oslo, Oslo, Norway

*Correspondence to*: Jan Magnusson (jan.magnusson@slf.ch)

**Abstract**

We present a high-resolution hydrometeorological and snow dataset from the alpine Dischma watershed and its surroundings in eastern Switzerland, including station measurements of variables such as snow depth and catchment runoff. This dataset is particularly suited for different modelling experiments using distributed and process-based models, including physics-based snow and hydrological models. Additionally, the data is highly useful for testing various snow data assimilation schemes and for developing models representing snow-forest interactions. The dataset covers seven water years from 1 October 2016 to 30 September 2023. The complete domain spans an area of 333 km² with altitudes ranging from 1250 to 3228 meters. The Dischma basin, with its outlet at 1671 m elevation, occupies 42.9 km². Included in the dataset are high-resolution (100 m) hourly meteorological data (air temperature, relative humidity, wind speed and direction, precipitation, as well as long- and shortwave radiation), land cover characteristics (primarily forest properties), and a digital elevation model. Noteworthy, the dataset includes snow depth acquisitions obtained from airborne lidar and photogrammetry surveys, constituting the most extensive spatial snow depth dataset in the European Alps. Along with these gridded datasets, we provide daily quality-controlled snow depth recordings from seven sites, biweekly snow water equivalent measurements from two locations, and hourly runoff and stream temperature observations for the Dischma watershed. The data compiled in this study will be useful for further developing our ability to forecast snow and hydrological conditions in high-alpine headwater catchments that are particularly sensitive to ongoing climate change.

## 1 Introduction

In this paper, we present a high-resolution hydrometeorological and snow dataset from the Dischma watershed and its surroundings in eastern Switzerland. The Dischma catchment represents a typical high-alpine watershed in the European Alps and serves as a headwater basin of the Rhine River. The Rhine flows through Switzerland, Germany, and the Netherlands before reaching the North Sea, passing through some of the most densely populated regions in Europe. Snow deficits in headwater catchments like Dischma can contribute to summer droughts far downstream, causing significant socio-economic



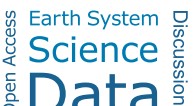

impacts (Avanzi et al., 2024). Conversely, rain-on-snow events can lead to devastating floods (Rössler et al., 2014). Additionally, snow cover influences ecosystem dynamics (Rixen et al., 2022), natural hazards (Ortner et al., 2023), tourism (Töglhofer et al., 2011), and climate through snow-albedo feedback (Thackeray et al., 2019). Therefore, accurately monitoring and reliably forecasting snow and water resources in these headwater catchments is of high importance for large range of applications.

To improve the predictability of water resources from mountainous regions, the International Network for Alpine Research Catchment Hydrology (INARCH) was initiated by Pomeroy et al. (2015) as part of the Global Energy and Water Exchanges project of the World Climate Research Programme. The Dischma catchment belongs to the study basins of the INARCH project (for more details, see https://inarch.usask.ca), and similar datasets as ours have already been presented for sites such as the Marmot Creek in Canada (Fang et al., 2019), Guadalfeo basin in Spain (Polo et al., 2019), Heihe catchment in China (Che

et al., 2019), Tuolumne and Merced watersheds in the USA (Roche et al., 2019), and the Rofental in Austria (Strasser et al., 2018). Our dataset, which features distributed meteorological forcing data and annual measurements of spatial snow depth distribution in the basin, will complement existing studies and provide the hydrological and snow modelling communities with valuable data for conducting simulation experiments for the high-alpine Dischma watershed.

Numerous snow and hydrological research studies have been conducted in the Dischma basin and its surrounding areas. Early
studies employed seminal methods for snowmelt runoff modelling (Kustas et al., 1994; Martinec, 1975; Rango and Martinec, 1981) and included assessments of groundwater hydrology (Martinec et al., 1982), the impact of avalanches on hydrology (Martinec, 2014), and the relationships between snow-covered area, snow water equivalent, and runoff volume (Martinec, 1982; Martinec and Rango, 1987). Data from the basin has been used to compare spatially distributed hydrological models in mountainous catchments, including simple energy-balance schemes for snow cover simulations (Gurtz et al., 2003; Zappa et
al., 2003), also in the context of climate change (Carletti et al., 2022).

The physics-based surface process model Alpine3D was initially tested using observations from Dischma (Lehning et al., 2006), with subsequent applications in this particular catchment for assessments of snow and climate change (Bavay et al., 2009; Bavay et al., 2013), runoff dynamics (Brauchli et al., 2017; Wever et al., 2017), and stream temperatures (Comola et al., 2015). Further studies in the catchment and its vicinity include analyses of avalanche activity (Baggi and Schweizer, 2009),
snow-forest interactions (e.g., Mazzotti et al., 2020b; Moeser et al., 2015), snow redistribution processes (Berg et al., 2024; Quéno et al., 2024), snow-atmosphere interactions (Mott et al., 2017; Haugeneder et al., 2024), and snowfall processes (Gerber et al., 2018; Reynolds et al., 2024). The Dischma catchment and its surroundings have been a focus area for developing cost-efficient techniques to acquire accurate, high-resolution snow depth data over large areas. This effort has resulted in a unique long-term dataset for the European Alps (Bührle et al., 2023).

The purpose of this paper is to provide a comprehensive high-resolution (100 m) meteorological and snow dataset along with relevant landcover data complemented by station measurements encompassing variables such as catchment runoff and snow water equivalent for the Dischma basin (area equal to 42.9 km$^2$) and its surroundings in eastern Switzerland (total area equal



to 333 km²). We provide hourly meteorological and runoff data over the complete study period from 1 October 2016 until 30
September 2023 spanning seven water years. The presented datasets serve as a pivotal resource for further improving and
testing distributed processes-based models including physics-based snow models, land-surface schemes and hydrological
models tailored for detailed simulations within high alpine regions. The dataset is processed to be directly useful in various
modelling experiments, and also suitable for testing various data assimilation schemes. We describe the different data sources
and processing methods employed, and where the data can be accessed below.

## 2    Site description – Dischma catchment and surroundings

The Dischma watershed, located in eastern Switzerland, spans an area of approximately 42.9 km² and serves as a headwater
basin for the Rhine River (Figure 1). The basin is situated in the transition zone between the wet northern Alps and the dry
central Alps (Carletti et al., 2022). The elevation within the catchment ranges from 1671 to 3146 m, with an average altitude
of 2371 m. Below, we provide an overview of the most significant catchment characteristics as described by Höge et al. (2023).
The mean slope of the catchment is 25°, with 83% of the basin area being steeper than 15°. Higher altitudes in the watershed
are primarily characterized by bare land with rocks and bare soil, covering 55% of the watershed area. At lower altitudes, grass
and herb vegetation dominate, accounting for 33% of the area. Coniferous forests cover 2%, and mixed forests cover 1% of
the basin. Wetlands, which make up 7% of the landscape, are distributed across the watershed, and the Scaletta Glacier,
occupying the southeasternmost part, covers 1% of the area. The glacier has an area of 0.49 km² and a volume of 0.01 km³,
according to the latest inventories. The soils in the Dischma watershed are mainly sandy, with smaller inclusions of silt, clay,
and minor amounts of organic material. Geologically, the basin is characterized by metamorphic rocks and unconsolidated
sediments. For more detailed information about the catchment attributes, refer to Höge et al. (2023) and the references therein.
Within the Dischma catchment, only one site with snow observations is available (Figure 1). To obtain a sufficient number of
locations with continuous snow observations for model evaluations, we expanded our domain and meteorological forcing
dataset (see Section 3) to cover an 18.5 by 18.0 km region surrounding the Dischma watershed. In this expanded region,
altitudes range from 1250 to 3228 m, with an average altitude of 2257 m. Additionally, this larger region covers lower
elevations with larger forest cover, important for conducting research on snow-forest interactions (e.g., Mazzotti et al., 2019b).
The snow observations are either performed by the WSL Institute for Snow and Avalanche Research SLF (SLF) or by the
Federal Office of Meteorology and Climatology (MeteoSwiss).

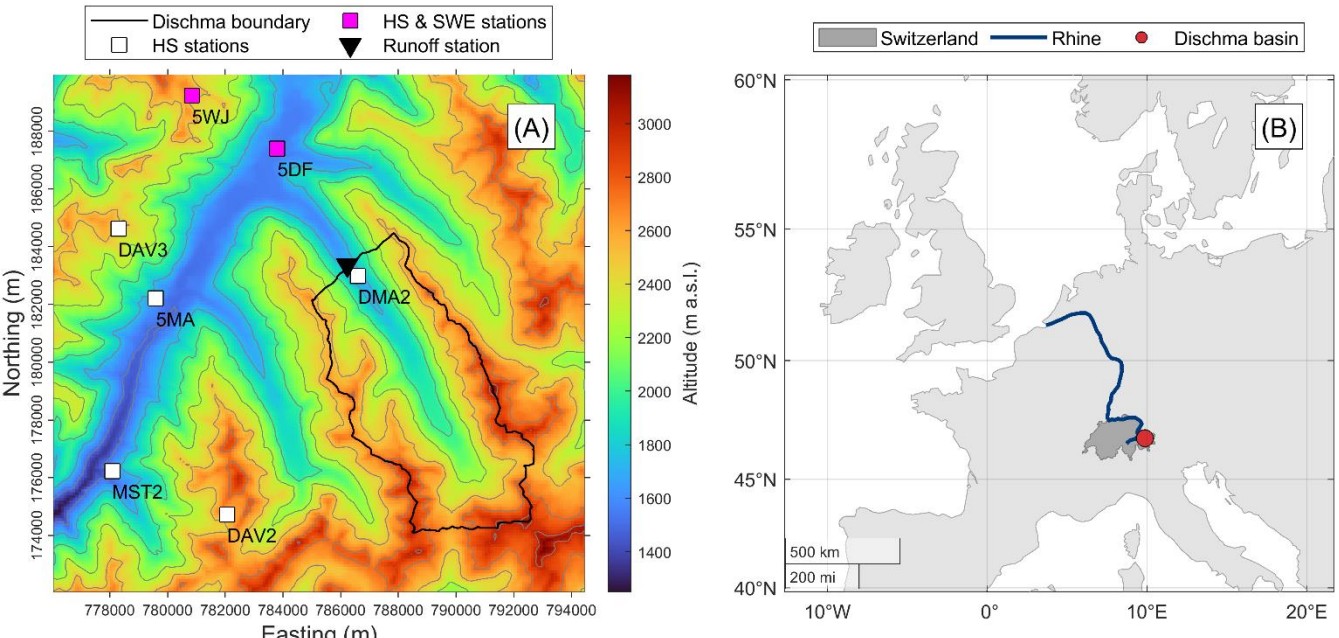

Figure 1. Site map of the Dischma watershed and extended domain for which we provide meteorological and land cover data (panel a). The map indicates the altitude of the domain as well as the location of stations used for snow, meteorological and runoff measurements. Location of the Dischma basin in Europe along with the Swiss national borders and the Rhine river (panel b).

## 3    Spatial meteorological data

Hourly gridded meteorological data (air temperature; relative humidity; long- and shortwave radiation; wind speed and direction; precipitation; air pressure) at 100 m resolution were derived from the analysis fields of the weather forecasting model COSMO and combined gauge and radar precipitation product CombiPrecip by employing variable-specific downscaling techniques as outlined below (see sections 3.1 to 3.4). Both the COSMO model results as well as CombiPrecip product are generated by MeteoSwiss. We provide the same forcing data as used by Quéno et al. (2024) for detailed snow cover simulations including wind- and gravity-driven snow redistribution. Equivalent meteorological datasets based on largely the same downscaling strategies have been used in various snow studies (e.g., Magnusson et al., 2017; Winstral et al., 2019). Importantly, the meteorological forcing data presented here but at 250 m resolution is used for real-time snow-hydrological forecasting in Switzerland (Mott et al., 2023). Thus, the high-resolution meteorological dataset presented in this paper is particularly valuable for assessing various model simulations under operational data constraints.

The COSMO (Consortium for Small-scale Modeling; see https://www.cosmo-model.org/) numerical weather prediction system was developed for regional-scale atmospheric simulations. COSMO employs a grid-nesting approach, enabling simulations at varying spatial resolutions, including 1.1 km in our case. As mentioned above, we use data from the analysis version of this model, which became operational in March 2016 (Winstral et al., 2019). The COSMO model incorporates a set



of primitive equations to describe atmospheric behaviour and uses a non-hydrostatic formulation suitable for depicting atmospheric processes at smaller scales and in mountainous regions such as the Swiss Alps. COSMO uses detailed

parameterizations for physical processes such as radiation, turbulence, and microphysics, enhancing its ability to capture a wide range of atmospheric phenomena. The non-hydrostatic formulation of the model allows for a better representation of features like meso-scale orographic effects, which are crucial for accurately capturing precipitation patterns for example (Vionnet2016). COSMO is widely used in operational weather forecasting and research applications, particularly for regions with complex terrain or small-scale weather features (Klasa et al., 2018).

In addition to data from the COSMO weather forecasting model, we also used precipitation fields from the hourly CombiPrecip product. These precipitation fields are derived using a geostatistical combination of heated rain-gauge observations and radar measurements, providing state-of-the-art spatial precipitation estimates. The product relies on data from automatic stations that record precipitation every 10 minutes and five C-band weather radars located within Switzerland. Within the study domain, there are two precipitation gauges with 10-minute measurements (near locations 5DF and 5WJ in Figure 1) and one radar (near

site 5WJ in Figure 1). A method based on co-kriging with external drift is used to combine the two data sources, resulting in nearly unbiased precipitation maps at an hourly resolution (Barton et al., 2020; Sideris et al., 2014).

Below, we describe the downscaling strategy for the various meteorological variables. Initially, we generated data at a 25 m resolution. Subsequently, to maintain reasonable data volumes, we upscaled the results to 100 m resolution for the extended data domain (see Figure 1) and aligned with the provided land cover data such as the digital elevation model (see section 7).

## 3.1     Air temperature and relative humidity

Air temperature was linearly interpolated onto the high-resolution grid, incorporating a lapse rate correction of 6.5 K/km, which was held constant over time. To obtain distributed relative humidity values, we first interpolated dew point temperatures onto the high-resolution grid using the same method and lapse rate as for air temperature. Finally, the interpolated air and dew point temperatures were converted to relative humidity using the saturation pressure function by Murray (1967). The obtained

fields represent air temperature and relative humidity at 10 m above ground.

## 3.2     Long- and shortwave radiation

Shortwave radiation was downscaled to account for terrain effects using the radiation transfer model HPEval (Jonas et al., 2020). HP masks direct shortwave radiation whenever surrounding terrain inhibits insolation, while diffuse shortwave radiation is scaled with the local sky view fraction (which is provided with this dataset, see section 7). The resulting incoming shortwave

radiation at the surface is output per inclined surface area. Incoming longwave radiation is provided as downwelling flux above terrain so that the influence of the surrounding terrain can be calculated inside the snow model based on the simulated surface temperatures. Note that incoming radiation below forest canopy should be treated using the forest radiation transfer parameters discussed in Section 4 below.





### 3.3 Wind speed and direction

To account for topographic effects on wind fields, wind speed and direction were downscaled using the mass-conserving dynamical downscaling model WindNinja (Forthofer et al., 2014; Wagenbrenner et al., 2016), version 3.7.0. The coarse resolution COSMO wind fields (see above) were used as forcing input to the model. Vionnet et al. (2021) and Quéno et al. (2024) noted model instabilities in very complex terrain when using the mass- and momentum-conserving version of WindNinja (Wagenbrenner et al., 2019), which was then not applied for the present domain. The final fields represent wind

speed and direction at 10 m above ground.

### 3.4 Precipitation

Hourly precipitation fields were obtained from the combined radar and rain-gauge product CombiPrecip (see description section 3). These precipitation fields were improved by assimilation of daily snow depth recordings using an optimal interpolation scheme. The combination of point snow depth recordings with the gridded precipitation field requires four steps.

(1) Daily snowfall rates were estimated at the snow monitoring sites using a density model (HS2SWE) that converts changes in snow depth to snow water equivalents. (2) Daily solid and liquid precipitation fields were obtained using the precipitation fields from CombiPrecip and gridded air temperatures from COSMO. (3) The daily gridded snowfall fields were improved by assimilating the snowfall estimates obtained at the stations in step 1 using an optimal interpolation scheme. (4) The daily improved snowfall fields together with the daily rainfall estimates were used to correct the hourly precipitation fields from

CombiPrecip. For detailed information about the assimilation procedure outlined here, see Magnusson et al. (2014), Mott et al. (2023) and Quéno et al. (2024).

### 3.5 Features of meteorological forcing fields

Figure 2 illustrates various features of the high-resolution meteorological forcing fields. A precipitation gradient is evident, with the north-eastern part of the region experiencing a wetter climate, while the south-eastern area is comparatively drier

(panel a). There is also a pronounced vertical gradient in precipitation, with valleys receiving less precipitation than mountain ridges. In the valleys, approximately 30 to 40% of the total precipitation falls as snow during the study period, whereas at higher altitudes, this fraction increases to roughly 70% (panel b). Over the Dischma catchment, approximately 60% of precipitation falls as snow. Throughout the study domain, the average air temperature ranges from -4.7 to 7.7 °C, reflecting the variations in altitude foremost (panel c). The coldest air temperatures are recorded in the south-eastern parts of the domain,

coinciding with the highest mountain peaks (compare with Figure 1a). In the valley floors, the average daily air temperature falls below zero degrees Celsius for roughly three months each year, whereas at higher elevations, negative daily average temperatures occur for approximately seven to eight months annually (panel d). Unlike the relatively smooth precipitation and air temperature fields, the downscaled shortwave radiation, which includes both direct and diffuse components, exhibits fine-grained features with significantly higher radiation inputs on south-facing compared to north-facing slopes (panel e). Similar



fine-grained patterns are observed in the downscaled wind fields, where average wind speeds exceed approximately 8 m/s along the ridges, while in lower and more sheltered areas, wind speeds are around 2 m/s in average (panel f).

**Figure 2. Panel a shows the average annual precipitation over the study period, while panel b illustrates the fraction of total precipitation that falls as snow. Panel c presents the mean air temperature during the study period, and panel d displays the fraction**
**of days with a daily average air temperature below zero degrees Celsius. Finally, panel e depicts the mean incoming shortwave radiation, and panel f visualises the average wind speed over the data period.**





## 4 Forest structure datasets

In addition to meteorological forcings, spatially distributed, process-based snow and hydrological models applied to (partially) forested domains require datasets characterizing the forest structure, as forest structure has a substantial impact on water and
energy exchanges between the atmosphere and the land surface. In this dataset, we have included forest structure descriptors used in the snow model FSM2oshd (Mott et al., 2023). As the forest representation in FSM2oshd was specifically developed to account for the impact heterogeneous forest structure (Mazzotti et al., 2021), it incorporates more forest structure variables than 'standard' land surface and hydrological models. Available datasets are described in the following sections. All these forest structure descriptors are derived from a 1 m canopy height model generated from airborne lidar datasets that cover all
of Switzerland (Webster et al., In prep - to be submitted soon). According to the model upscaling strategy proposed in Mazzotti et al. (2021), forest structure metrics are calculated at very high resolution (here: evaluation points at 10 m spacing) and then aggregated to 100 m to match the resolution of the meteorological datasets.

### 4.1 Canopy structure variables

Canopy structure descriptors provided in this study include the following variables: canopy height (Hc), local and stand-scale
canopy cover fraction (Fcl and Fcs), and leaf area index (LAI). These descriptors are valid for the fraction of the grid cell covered by forest, which is also provided in addition to the parameters listed above (Ffor). These variables are required as land use information by FSM2oshd, while many land surface and hydrological models require LAI and canopy height only. Hc, Fcl and Fcs are computed from the 1 m nationwide canopy height model (CHM), over areas of 5 m (Fcs) and 50 m (Hc, Fcs) radii around each evaluation point. LAI is parameterized from Fcl and Hc, with maximum LAI values limited to specific values for
forest type (needleleaf vs broadleaf, evergreen vs deciduous) and eco-regions and assuming leaf-off wintertime conditions. As these datasets were specifically created for snow modelling, seasonal variations in LAI and forest cover are disregarded. For further detail, we refer to Mott et al. (2023).

### 4.2 Radiation transmission variables

Sky-view fraction (Vf) and time-varying direct-beam transmissivity (Td(t)) are used to calculated the transmission of diffuse
and direct shortwave radiation, respectively, as well as calculate longwave radiation (Vf only). Both are dimensionless variables that describe the proportion of above-canopy radiation transmitted through the canopy to the snow/ground surface. In particuar, transmission of direct solar radiation through forest canopies is a highly complex process that is dependent on the position of individual tree crowns relative to the sun-position. Spatially and temporally, it varies over meter and minute scales. Vf and Td(t) are provided as input to FSM2oshd and therefore included in the datasets provided in this study. In a snow
modelling context, this allows a detailed representation of canopy radiative transfer without added model complexity and has been shown to be beneficial for capturing the impact of complex canopy structure on forest snow dynamics. The radiation transfer information has proven useful also in the context of ecological modelling (Zellweger et al., 2024).



Both Vf and Td(t) were calculated using the Canopy Radiation Model (CanRad; Webster et al., 2023; Webster et al., In prep -
to be submitted soon). CanRad calculates synthetic hemispheric images at each evaluation point, first by tracing the canopy
horizon line from the CHM, then calculating a probability of light transmission below this line using a statistical relationship
between canopy thickness and tree crown volume, which is tree-type dependent across the model domain, determined using
the forest mix rate data from Waser et al. (2017). Vf and Td(d) are calculated using the methods in Essery et al. (2008) and
Jonas et al. (2020), respectively. The 10 m evaluation points used in this study are part of the SwissRad10 dataset (Webster et
al., In prep - to be submitted soon), which were aggregated to 100 m for FSM2oshd. In this dataset, both Vf and Td(t) are
calculated assuming leaf-off canopy conditions. Like the canopy structure variables (Section 4.1), Vf and Td(t) relate to the
forest fraction of each grid cell.

## 5   Snow data

### 5.1   Spatial snow depth data from airborne surveys

High-resolution snow depth (HS) data for the Dischma basin and surrounding areas were obtained from airborne
photogrammetric and lidar surveys. The photogrammetric surveys were conducted once each winter (from mid-March to mid-
April) throughout the entire study period, while the three lidar acquisitions were collected during the winter-to-spring transition
in 2017. It is important to note that lidar data provide measurements in both forested and open areas, whereas the
photogrammetric method only captures data in open areas. The lidar surveys have been used in studies of forest-snow
interactions (Mazzotti et al., 2019b; Mazzotti et al., 2023; Yang et al., 2023; John et al., 2022), for validation of snow modelling
including snow redistribution processes (Berg et al., 2024; Quéno et al., 2024), and for spatially distributed modelling of snow
instability (Richter et al., 2021). Photogrammetric HS data obtained within the study domain has been applied in snow depth
modelling and mapping studies as key validation and training datasets (e.g., Helbig et al., 2021; Daudt et al., 2023). Notably,
these large-scale and high-resolution acquisitions of snow depths are unique for the European Alps. Those datasets are
comparable in quality to the results generated with lidar altimetry by the Airborne Snow Observatory (ASO) program in the
United States (Painter et al., 2016; Meyer et al., 2022), but the monitoring started earlier, in 2010 (Bühler et al., 2015). The
Dischma acquisitions represent the longest data record of large-scale and high-resolution HS monitoring based on airborne
remote sensing, however, with a smaller area than covered by the ASO program.

The photogrammetric HS fields provided in our dataset were acquired using the survey grade frame camera UltraCam
providing data with a spatial resolution of 0.5 m (Bührle et al., 2023). These datasets have been compared to manual
measurements and drone-based photogrammetry (Vander Jagt et al., 2015; Bühler et al., 2016) as well as results from satellite
photogrammetry (Marti et al., 2016), resulting in a HS accuracy in terms of root-mean-squared-error ranging from 0.1 to 0.15
m (Eberhard et al., 2021). The three lidar surveys were performed on 20 March 2017, 31 March 2017, and 17 May 2017 over



a region centred on the Dischma valley, and the resulting data has a spatial resolution of 1 m. Mazzotti et al. (2019b) validated
these datasets in sparse forested terrain against roughly 11000 measurements of HS taken by manual probing, showing a low
bias (-4 to 0 cm) and root-mean-square error (4 to 8 cm) between the two datasets. For both the photogrammetry and lidar
datasets, urban areas (e.g., buildings) and outliers were masked out. Additionally, forested terrain was excluded from the
photogrammetric data. These filtering steps enhance the suitability of the datasets for model evaluation purposes.

In our dataset, we have upscaled (averaged) the high-resolution HS data from both photogrammetry and lidar to the common
100 m grid used for all spatial data (e.g., land cover and meteorological fields). Thus, the observed HS fields can directly be
used for evaluating model simulations performed with the provided meteorological forcing data (see section 3). The native-
resolution photogrammetry data can be obtained from Bührle et al. (2022). Overall, the upscaled lidar acquisitions cover an
area ranging from 139 to 150 km$^2$, or 42 to 45% of the complete domain. Excluding 2018, the upscaled photogrammetric data
covers an area ranging from 165 to 284 km$^2$, or 50 to 85% of the domain. In 2018, bad weather and sensors failures resulted
in smaller acquisition area equal to 51 km$^2$, or 15% of the domain. Overall, this dataset provides a unique opportunity for
evaluating physics-based snow models including redistribution processes in the European Alps (e.g., Quéno et al., 2024).

Figure 3a illustrates a HS field obtained through photogrammetry for the Dischma valley and its surroundings. The method
does not provide valid data outside the acquisition perimeter and in low-elevation areas with forests and buildings, so these
regions area masked out and marked in grey (as described above). The largest snow accumulations are observed in the south-
eastern and high-altitude areas (see also Figure 1), even though this area belongs to the drier part of the domain according to
our precipitation fields (see Figure 2a). These large snow accumulations align with the small glacier located along the southern
border of the Dischma catchment (see section 2).

Figure 3b presents simulation results for the same domain and date, obtained using the FSM2trans model, which includes snow
redistribution by wind and avalanches (Quéno et al., 2024). The model was run with the datasets presented in this study.
Overall, the model captures the main features of the observed snow cover reasonably well, such as (a) the strong elevation
trend in HS, (b) the large snow accumulations due to snowdrift and, below steep slopes, due to avalanches, and (c) the snow-
free ridges caused by wind-erosion (although sometimes erosion is too strong). At the same time, at the highest altitudes above
2500 m, the model displays lower snow depths than the observations, likely due to underestimated precipitation (Quéno et al.,
2024). The model also fails to capture some of the small-scale variability in the observed HS, in particular at mid-altitudes
(compare Figure 1a and b). Despite these limitations, this example demonstrates that valuable model results can be obtained
from the data provided in this paper. Such results can lead to interesting conclusions and potentially improve both the forcing
data (e.g., better correction of precipitation data) and the models (e.g., more accurate wind redistribution schemes).





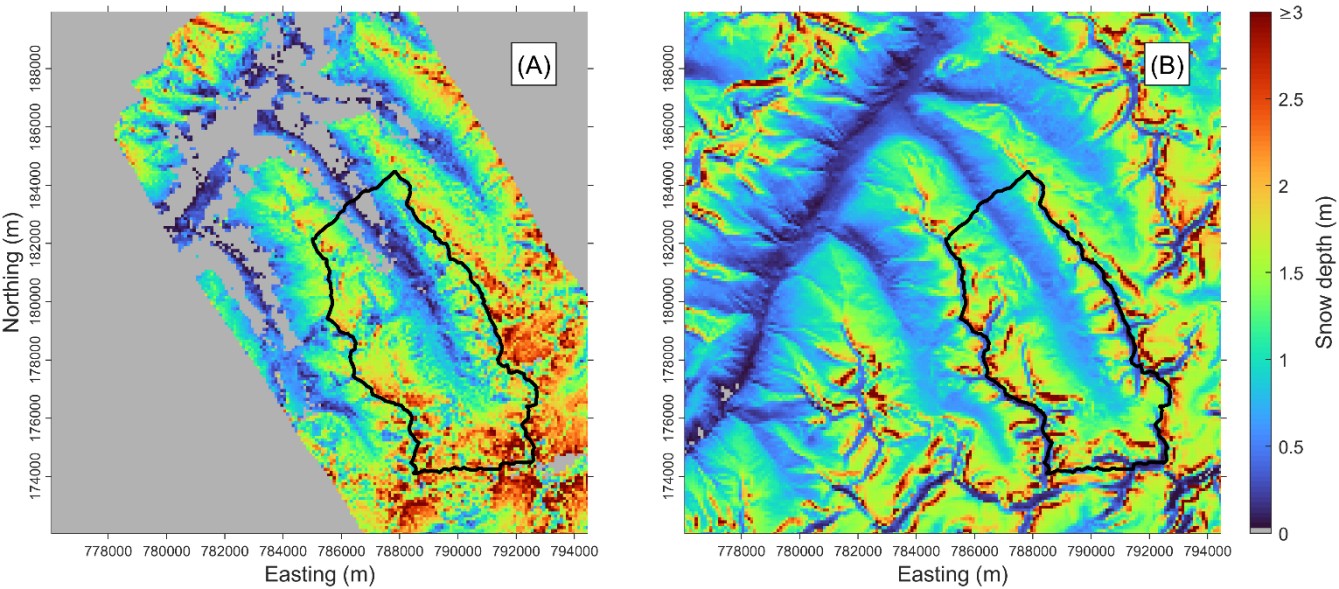

**Figure 3. Observed (panel a) and simulated (panel b) snow cover at 100 m resolution over the study domain for 6 April 2020. Areas**
**without observations are masked by grey colour in panel a.**

### 5.2 Station snow depth measurements

Daily HS recordings from 7 stations and biweekly snow water equivalent (SWE) measurements from 2 sites were provided in this dataset (Table 1). All measurement locations are characterized by limited exposure to wind and flat terrain. The HS observations were performed either manually using a fixed installed and graded measurement stake or automatically using

ultrasonic sensors. The manual daily HS recordings have an uncertainty in the range of ±2 cm (Resch et al., 2023), while the corresponding uncertainty for the automatic measurement is in the order of ±1 cm (Ryan et al., 2008). The HS recordings provided in this dataset have been quality checked manually. Gaps have been filled and outliers have been corrected through interpolation procedures utilizing data from surrounding stations. These quality-controlled daily HS data are routinely used within the snow-hydrological forecasting service operated by SLF (Mott et al., 2023).

The biweekly SWE observations were obtained through manual profiling by taking vertical snow cores from the top to the bottom of the snowpack with the ETH sampler. The ETH cylinder has demonstrated lower variability between sample replicates compared to most other devices used for measuring SWE in Europe and North America (López-Moreno et al., 2020). Each SWE observation is paired with an HS recording from the same snow pit. Note that the HS measurement from the pit may differ from the daily HS recording taken at the nearby stake or automatic sensor due to spatial variability in the snow

cover. The ratio between the SWE and HS observation obtained from the manual profile gives an estimate of the bulk snow density. The manual snow profiles are conducted by experienced field staff and the data is used operationally by the avalanche forecasting service as well as by the snow-hydrological forecasting team at SLF.



**Table 1. Sites with daily snow depth recordings and biweekly snow profiles providing SWE estimates. Data providers are either MeteoSwiss (MCH) or SLF. The DAV2 and DAV3 sites are equipped with automatic sensors, while at the remaining locations the measurements are performed manually.**

| Name | Altitude (m) | Daily HS | Biweekly SWE/HS |
|---|---|---|---|
| Dischma (MCH.DMA2) | 1710 | x | |
| Monstein (MCH.MST2) | 1575 | x | |
| Davos Flüelastr. (SLF.5DF) | 1560 | x | x |
| Matta Frauenkirch (SLF.5MA) | 1655 | x | |
| Weissfluhjoch (SLF.5WJ) | 2540 | x | x |
| Bärentälli (SLF.DAV2) | 2558 | x | |
| Hanengretji (SLF.DAV3) | 2455 | x | |

Figure 4 shows HS and SWE recordings from the stations (see Figure 1 for their locations). The seasonal snow at low-altitude stations, such as SLF.5DF, typically lasts for about five months, from November to March (see Figure 4a). At higher-altitude locations, like the SLF.5WJ site, the snow cover usually extends for around eight months, from November to June (see Figure 4b). Peak snow depths at the low-altitude SLF.5DF site ranges between 0.46 and 1.75 m, while at the high-altitude SLF.5WJ site, they range between 1.98 and 3.13 m. Overall, the high-altitude location shows 79% to 330% higher peak HS than the low-altitude site. For SWE, maximum values range between 110 and 477 mm at SLF.5DF, whereas at SLF.5WJ, they range between 636 and 1389 mm. Consequently, the high-altitude site exhibits approximately 117% to 478% higher peak SWE values than the low-altitude location. Thus, the station recordings display significant difference in snow thickness and mass between low and high altitudes, typical for the European Alps.

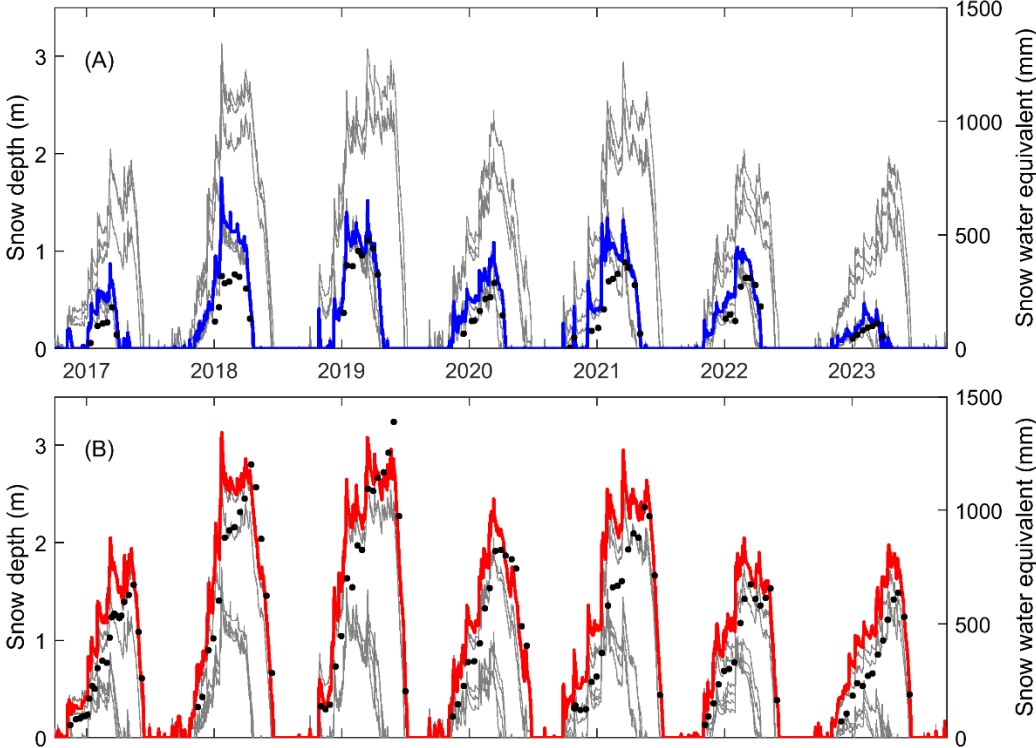

**Figure 4. Snow depth observations from the low-elevation location SLF.5DF (blue line in panel a) and the high-alpine site SLF.5WJ (red panel in panel b) together with biweekly SWE measurements (black dots). The grey lines represent snow depth recordings from the remaining stations within the study domain (see also Table 1 and Figure 1).**

## 6    Hydrological data

Runoff and temperature in the Dischma stream are measured at the Kriegsmatte station, operated by the Federal Office for the Environment (FOEN). This station, located at 1671 m.a.s.l., has been recording discharge since 1963 and stream temperature since 2004. Figure 1 displays the location of this runoff gauging site, which has identifier 2327 according to FOEN terminology. In this dataset, we provide hourly averages of the recorded flows and temperatures in the Dischma stream for the complete study period. Below we present various long-term statistics of the temperature and stream flow recordings, that were acquired from FOEN (see hydrodaten.admin.ch, last accessed 27 May 2024).

Figure 5 shows various characteristic of the stream measurements in relation to the meteorological data included in this dataset (see section 3). The variations in monthly average air temperatures for the Dischma catchment are larger than the variations in the stream temperatures (see Figure 5a). The monthly mean water temperature varies between 0.5 and 9.9 °C, whereas the corresponding monthly mean air temperatures range from -11.7 to 10.1 °C. Over the complete study period, the average water temperature equals approximately 4.4 °C, which is very close to the long-term average of 4.3 °C.

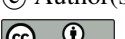



The discharge in the Dischma stream is heavily influenced by snowmelt, resulting in high stream flows in May and June, whereas the winter period is marked by low flows (see Figure 5b). The average discharge measured over the study period equals approximately 1.7 m³/s, matching the long-term average precisely. Runoff exceeding 11.4 m³/s has a return period of 2 years, while runoff surpassing 19.5 m³/s has a return period of 100 years. Low flows are characterized by a return period of 2 years for discharges below 0.31 m³/s, and a return period of 100 years for runoff below 0.12 m³/s. Due to a significant portion of precipitation falling as snow (see Figure 2b), stream flow is largely disconnected from precipitation events, and only occasionally reacts to rainfall episodes.

The variations in annual total runoff align well with the cumulative precipitation sums over the Dischma catchment (Figure 5c). The runoff ratio, defined as stream flow divided by precipitation, ranges from 0.91 to 1.07 over the seven-year period. In two of those years, the runoff ratio exceeds one, suggesting that either the rainfall and/or snowfall amounts are underestimated, even after applying our corrections outlined in section 3.4. Annual evapotranspiration amounts to approximately 260 mm in Dischma according to model-derived estimates provided by Höge et al. (2023), suggesting that the runoff ratio should equal approximately 0.82. Thus, likely, our precipitation forcing data still underestimates actual precipitation over the complete study period, despite our corrected precipitation fields showing significant improvements compared to other commonly used precipitation products in Switzerland. For instance, RhiresD, also provided by MeteoSwiss, entails runoff ratios of approximately 1.25 for the Dischma catchment in the CAMELS-CH dataset (Höge et al., 2023) and consequently shows a much stronger underestimation of actual precipitation than our dataset.


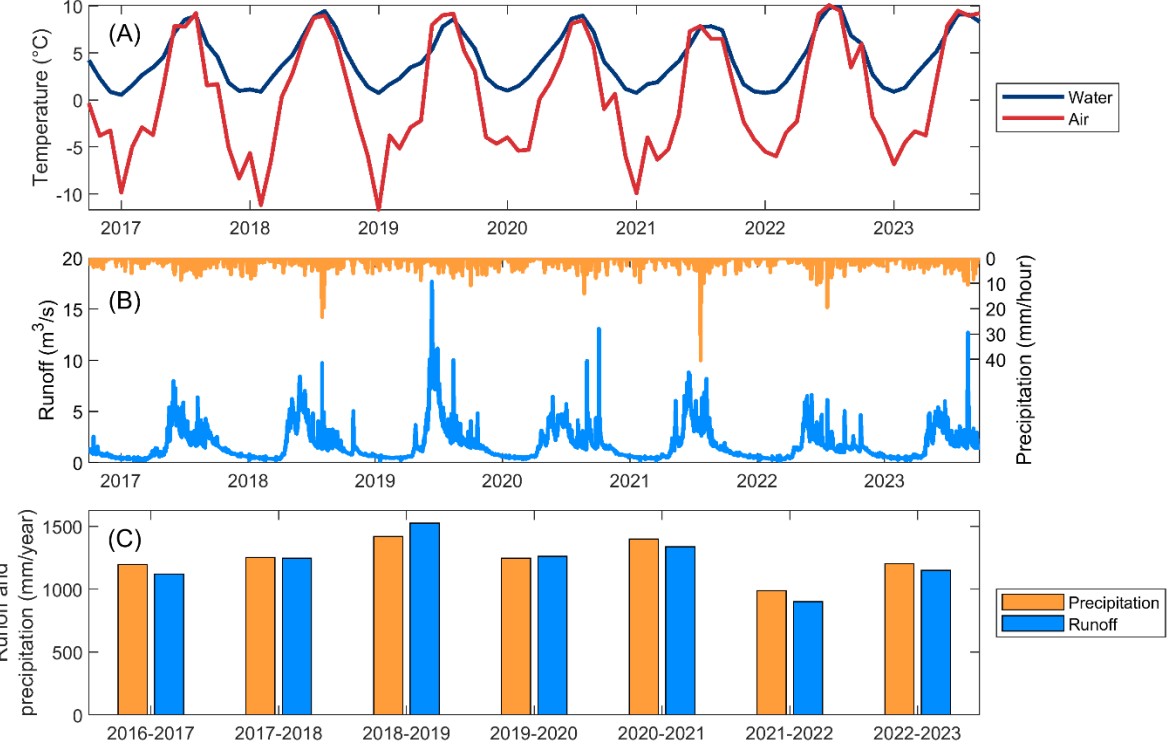


**Figure 5. Average monthly water temperature measured at the Kriegsmatte station in Dischma along with catchment-averaged monthly air temperature (panel a). Measured hourly runoff together with basin-averaged hourly precipitation (panel b). Yearly sums, from first of October to last of September, of measured runoff and precipitation averaged over the catchment area (panel c).**

## 7    Additional data describing the Dischma catchment and surroundings

Along with the data presented above, we provide a digital elevation model at 100 m resolution that was computed by upscaling the 25 m model from Federal Office of Topography (swisstopo) in Switzerland, and a polygon defining the area draining to the Kriegsmatte runoff station. All gridded data (e.g., land and forest cover, spatial snow depth maps, and meteorological forcing data) were aligned with this digital elevation model.

In addition to the data provided in this paper, various open data sources may be relevant for different modelling applications,

many of which are referenced to in the CAMELS-CH dataset (Höge et al., 2023). For instance, land cover and land use information can be sourced from the Coordination of Information on the Environment (CORINE) dataset (Büttner et al., 2004), which was last updated in 2018. This dataset offers a land inventory in 44 thematic classes (e.g., natural grassland, bare rock, and the combined class glaciers and perpetual snow) with a spatial resolution of 100 m or better. Additionally, the 3D soil hydraulic database of Europe, with a resolution of 250 m, provides data on variables such as saturated water content and

saturated hydraulic conductivity at seven levels up to a soil depth of 2 m (Tóth et al., 2017). The European Soil Database Derived data offers further insights into soil composition (Hiederer, 2013a, b). Geological information can be accessed through

the high-resolution global lithology map, which categorizes rock types into a limited set of classes (Hartmann and Moosdorf, 2012). Finally, the Swiss Glacier Inventory contains detailed glacier data, including areas, outlines, and debris cover of all glaciers in Switzerland (Linsbauer et al., 2021). The above-mentioned open data sources are among the most significant

references within the CAMELS-CH dataset (Höge et al., 2023) and can potentially be useful in various modelling applications related to the data presented in this study.

## 8    Data availability

The data presented in this paper is available at EnviDat (https://drive.switch.ch/index.php/s/ofy2ZW4yVH7dhET; temporary URL used during the review process). For all files, we use the Swiss coordinates system CH1903/LV03 (EPSG:21781) while

all date information is given in Central European Time (UTC+01:00). The meteorological forcing data is stored as self-explanatory NetCDF-files formatted to produce correct visualizations in software packages such as Panoply (https://www.giss.nasa.gov/tools/panoply; last access, 20 June 2024) and the files adhere to standard NetCDF conventions (see global attributes of respective files). Each NetCDF file contains a set of three-dimensional arrays containing one day of hourly meteorological data for each meteorological variable (see section 3). The time-varying direct-beam transmissivity data (see

section 4.2) is stored in a single NetCDF file. The station measurements (e.g., runoff and snow water equivalent) are stored in comma separated text files with headers containing all necessary metadata (e.g., station coordinates). All raster files (e.g., spatial snow depth maps and forest cover data) are stored in GeoTIFF format.

Several complementary datasets have been published on EnviDat (https://www.envidat.ch/) that may be relevant for evaluating modelling experiments performed with the data published in this paper. Examples of particular interest include forest canopy

structure data for radiation and snow modelling (Mazzotti et al., 2020a), snow and canopy data for high-resolution forest snow modelling (Mazzotti et al., 2019a), and input datasets for forest snow modelling in Flüela valley northeast of Dischma (Mazzotti and Jonas, 2022). The Flüela valley is within the domain of the datasets presented in this paper. Additionally, drone based snow depth maps, 10 in total, with a spatial resolution of 0.1 m and an accuracy of approximately 0.1 m are available for a 4 km$^2$ subset of the Dischma catchment in the area of the Schürlialp located in the southern part of the basin (for data,

see Bühler et al., 2022). These datasets have an extremely high level of detail and were applied as reference data for accuracy assessment of the photogrammetric datasets (Bühler et al., 2017; Eberhard et al., 2021). Additionally, data from the well-known Stillberg ecological treeline research site established 1975 in Dischma is also freely available (Lechler et al., 2024). Finally, a large range of datasets from the Dischma valley and surrounding areas such as biweekly snow profiles from Weissfluhjoch and glide-snow avalanche activity at Dorfberg north of Dischma can be obtained from EnviDat.

## 9  Summary and Outlook

We provide seven water years of high-resolution (100 m) spatially distributed hourly meteorological data for the Dischma catchment and surrounding areas in eastern Switzerland useful as input to various models. This model forcing data includes air temperature, relative humidity, wind speed and direction, short- and longwave radiation, and precipitation, alongside land and forest cover information. The model forcing data, used by Quéno et al. (2024) for detailed snow simulations including snow redistribution processes, have been formatted for ease of use to facilitate different modelling experiments. We additionally provide forest structure data, based on a detailed canopy height model, enabling modelling of forest-snow interactions in heterogeneous landscape, thus ensuring that the impact of forest cover is accurately captured in snow and hydrological applications.

In addition to the model input data, we have assembled validation datasets, including: (a) snow depth measurements from airborne lidar and photogrammetry surveys, quality-controlled and upscaled to the 100 m grid, (b) daily controlled and gap-filled snow depth recordings, as well as biweekly snow water equivalent observations, both from ground locations, and (c) hourly runoff and stream temperature measurements at the basin outlet of the Dischma catchment. The lidar and photogrammetric surveys provide the most extensive spatial snow depth dataset in the European Alps, with comparable quality to the Airborne Snow Observatory in the United States (Painter et al., 2016) but covering a smaller area.

Overall, the dataset is valuable for enhancing physics-based snow, land-surface, and hydrological models with applications in high-alpine catchments. This dataset is particularly useful for modelling snow redistribution through wind and avalanches, snow-forest interactions, and developing snow data assimilation schemes based on forcing data typically available in an operational setting. We anticipate that this dataset will lower the barriers for the snow and hydrological modelling communities to conduct simulation experiments in the Dischma region, a typical headwater catchment in the European Alps with a long history of diverse research studies.

## 10  Author contribution

J.M. and T.J. initiated and coordinated the study and processed the meteorological data. Y.B., L.Q., and B.C. compiled and processed the spatial snow depth data, while G.M. and C.W. handled the forest cover data. R.M. was responsible for processing the land cover data. J.M. also compiled the remaining datasets, such as runoff observations, wrote the manuscript, and created all the figures. All authors contributed to the analysis and supported the writing of the manuscript.

## 11  Competing interests

The contact author has declared that none of the authors has any competing interests.



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
