# Peer review of "High-resolution hydrometeorological and snow data for the Dischma catchment in Switzerland"

_Earth System Science Data, 2024_

## Author Response (AR1)

**Comment from editor:**

Dear Authors,

Many thanks for your manuscript, and for your intention to share these potentially valuable data for subsequent (re)use.

I am happy to begin the peer-review and discussion process.

Without going beyond the scope of a data description, I was wondering whether you add a short section on the catchment's water balance? I understand that there is some discrepancy (as yet unresolved?) between annualised precipitation and discharge. If so, it could be useful to alert potential use users to this. If you agree, this could be added during any eventual revisions.

Best regards,

James Thornton

**Reply to editor:**

Dear Editor,

The original manuscript already contained a section describing observed discrepancies in the water balance (see lines 325-334 in the initially submitted manuscript). To make this information more visible, we have (a) adapted the title of original section from "Hydrological data" to "Hydrological data and water balance analysis", and (b) added a sentence noting that further corrections of the precipitation data may be necessary in hydrological modelling applications. See lines 318 and 346-347 in the revised manuscript for these changes.

Best regards,

Jan Magnusson

**Reply letters to reviews for manuscript "High-resolution hydrometeorological and snow data for the Dischma catchment in Switzerland" by Magnusson et al.**
* * *
**Reviewer 1**

**Reviewer:** I read with pleasure the very nice manuscript by Magnusson et al. on data from the Dischma catchment in Switzerland. This is one of the most important research catchments in snow hydrology in Europe, and the manuscript is a very welcome addition to the existing literature at it outlines and delivers a hydrologically complete dataset to pursue snow hydrology science using data from this catchment.

I have only some very minor comments and recommend the manuscript to undergo a round of minor revision.

**Authors:** Thank you for your positive feedback on our study and your valuable comments for improving the manuscript. Below, we have provided our responses to your comments and outlined the changes we have made to the paper.
* * *
**Reviewer:** Abstract, line 9: may be worth starting by mentioning the exact spatial / temporal resolution rather than saying "high resolution" (as later done at line 15).

**Authors:** We have added the requested information to the abstract.

**Changes:** L 14.
* * *
**Reviewer:** line 18: "the most extensive spatial snow depth dataset": I guess you mean from lidar and/or photogrammetry correct? This does not include reanalyses or satellite observations. Perhaps it would be good to mention this by simply saying "the most extensive spatial snow depth dataset derived using such techniques" (as you already mention lidar and photogrammetry before)

**Authors:** Added.

**Changes:** L 24-25.
* * *
**Reviewer:** line 79: mention which is the latest inventory used?

**Authors:** The inventory is based on "Glacier Inventory 2016" described in Linsbauer et al. (2021). We have added this information to the manuscript.

Linsbauer, A., Huss, M., Hodel, E., Bauder, A., Fischer, M., Weidmann, Y., Bärtschi, H. & Schmassmann, E. 2021, The new Swiss Glacier Inventory SGI2016: From a topographical to a glaciological dataset. Frontiers in Earth Science, 22, doi:10.3389/feart.2021.704189.

**Changes:** L 87.
* * *
**Reviewer:** Section 3.1: I was a bit surprised to see a constant temperature lapse rate here. One could consider at least seasonal or monthly values. Why was this choice made?

**Authors:** The elevation difference between the 1.1 km COSMO grid and the 100 m grid, to which we downscale the weather forecasting model data, is less than 85 m for 50% of the 100 m grid cells and less than 203 m for 90% of the cells. Seasonal lapse rates in the European Alps, particularly in the nearby Italian and Austrian Tyrol regions, vary from 4.5 K/km (December–January) in winter to 6.5 K/km (April–August) in summer, as reported by Rolland (2003). Based on these variations and our assumption of a constant lapse rate of 6.5 K/km, combined with the elevation differences described above, the error introduced compared to using a seasonally varying lapse rate is estimated to be less than 0.2 K for 50% of the grid cells and less than 0.4 K for 90% of the grid cells during the coldest months (December–January). During the remaining months, the estimated errors are typically much lower. Considering these findings in light of other uncertainties, such as those associated with precipitation, we find the use of a constant lapse rate for temperature downscaling to be reasonable.

Rolland, C., 2003: Spatial and Seasonal Variations of Air Temperature Lapse Rates in Alpine Regions. J. Climate, 16, 1032–1046, https://doi.org/10.1175/1520-0442(2003)016<1032:SASVOA>2.0.CO;2.
* * *
**Reviewer:** Section 3.4: what do you mean with "optimal assimilation scheme"? Also, I am a bit puzzled by the fact that all weather variables but precipitation are from COSMO, while precipitation comes from CombiPrecip. How is correlation and consistency between precipitation and other variables (e.g., relative humidity or incoming shortwave radiation) preserved?

**Authors:** We utilize an "optimal interpolation scheme" to assimilate ground snowfall data, a widely used data assimilation method for precipitation analysis. We have included citations in the manuscript to clarify that "optimal interpolation" refers to a specific data assimilation technique.

According to MeteoSwiss, CombiPrecip "provides the best estimate of ground-level precipitation distribution currently available for Switzerland" (https://www.meteoswiss.admin.ch/services-and-publications/service/weather-and-climate-products/combiprecip.html; last accessed 2024-11-19). For this reason, we selected CombiPrecip over the precipitation fields generated by COSMO. At the same time, COSMO incorporates the same radar data as CombiPrecip in its analysis by applying an approach known as latent heat nudging (Leuenberger, 2005). This technique adjusts atmospheric thermodynamic quantities to align predicted precipitation rates from COSMO with raw radar estimates. However, unlike CombiPrecip, latent heat nudging does not incorporate ground-level precipitation measurements. To summarize, the latent heat nudging scheme reduces differences between COSMO and CombPrecip precipitation estimates. This leads to much reduced inconsistencies between precipitation given by CombiPrecip and other variables obtained from COSMO (e.g., relative humidity and shortwave radiation).

Leuenberger, D., 2005: High-Resolution Radar Rainfall Assimilation: Exploratory Studies with Latent Heat Nudging. Diss. ETH No. 15884, Research Collection, http://hdl.handle.net/20.500.11850/48174.

**Changes:** L 160-161.

**Reviewer:** line 182: "of" after "impact"?

**Authors:** Changed.

**Changes:** L 194.
* * *
**Reviewer:** line 264: isn't this underestimation of precip a bit in contradiction with the previous statement of CombiPrecip providing unbiased hourly precip fields (line 121)? I am not surprised about the potential underestimation at high elevations, so perhaps mention this in the description of CombiPrecip too (see also the discussion about the runoff ratio later)?

**Authors:** The evaluation of CombiPrecip was performed using precipitation gauges located on altitudes mainly below 2000 m.a.s.l. Thus, the quality of the precipitation product at high altitudes is more uncertain, while at lower altitudes the verifications show low biases. For better clarity, we have added a sentence informing that the evaluation of CombiPrecip was made using precipitation measurements with the majority located on altitudes below 2000 m.a.s.l.

**Changes:** L 130-132.

**Reviewer:** This is a review of "High-resolution hydrometeorological and snow data for the Dischma catchment in Switzerland". Overall, this is a very well written manuscript. The data are well described, and other than a few points of clarity that need to be added, it is in good shape.

**Authors:** Thank you for your positive feedback on our study and your valuable comments for improving the manuscript as well as the dataset. Below, we have provided our responses to your comments and outlined the changes we have made to the paper.
* * *
**Reviewer:** My main issue was getting the zip file decompressed. It seems to require a zip64 compliant decompressor (e.g., 7z) as Macos' Finder or CLI `unzip` could extract the files. I recommend that the authors either explicitly note that the file is in zip64 and requires a decompression program that can handle this format. Or, use a different format (tar.gz). I would also recommend the authors.

**Authors:** We have added information that the files were compressed using Zip64.

**Changes:** L 384-385.
* * *
**Reviewer:** The met netcdf time units appear to be incorrect

    double time(time) ;

        time:standard_name = "time" ;

        time:units = "days since 2016-10-01T00:00:00+01:00" ;

I believe this should be "hours since".

**Authors:** The time units "days since…" are correct in the files for the meteorological data. For example, this is the output (in split days) when reading the time variable for the first file in the dataset using Matlab:

time = ncread("METEO_DATA_201610160000.NC","time");

ans = [0, 0.0417, 0.0833, 0.1250, 0.1667, 0.2083, 0.2500, 0.2917, …, 0.9583]
* * *
**Reviewer:** Neither Panoply nor Paraview (both CF compliant nc loaders) correctly load the time component and cannot produce a xy-time visualization. I think it is because although it is not required, CF recommends Time x Z x Y x X and the data here have time dim last. https://cfconventions.org/Data/cf-conventions/cf-conventions-1.11/cf-conventions.html#dimensions

**Authors:** Unfortunately, we cannot reproduce the behaviour reported by the reviewer. For us, Panoply Version 5.3.4 correctly visualizes the gridded meteorological data on Windows 11 (see Figure

1 below for an example).  We have added information about Panoply version and operating system to the manuscript.

[Figure]

Figure 1. Screen shot showing air temperature for 2020-01-25 at 11:00 using Panoply from the METEO_DATA_202001250000.NC file in the provided dataset.

**Changes:** L 377.
* * *
**Reviewer:** The CF standards page lists the incoming fluxes (e.g., downwelling_longwave_flux_in_air) to have a canonical unit of W m-2 – is there a reason the authors deviate from this?

**Authors:** We have updated the canonical unit to the correct standard (W m-2) in the final dataset. The updated dataset can be accessed using the following link during the review process: https://wslch365-my.sharepoint.com/:u:/g/personal/jan_magnusson_slf_ch/Ed4CWKkJEG9JqyMdla8s5dEBwlnGVfiIsin sh8LDM3yOcg?e=qnJP9Q
* * *
**Reviewer:** The GeoTIFFs are missing a no-data value. It is presumed to be -9999 but this should be explicitly set, e.g.,

```

gdal_edit.py -a_nodata

```

**Authors:** We have updated the GeoTIFF files in the final dataset to explicitly include a no-data value.
* * *
**Reviewer:** L15: here and throughout, the use of a debiased NWP output needs to be clearly noted to be NWP output and not observations.

**Authors:** Clarified.

**Changes:** L 22, 70-71, 401-402.
* * *
**Reviewer:** L60 note hourly met data

**Authors:** Added.

**Changes:** L 66-67.
* * *
**Reviewer:** L70 throughout this section with respect to the percentages of land cover: it is not clear if the authors are describing the sub-set area, or the entire basin. For example, on L76 "accounting for 33% of the area" it is not clear if it is 33% of the lower elevations or of the total basin. Please clarify this throughout

**Authors:** Clarified.

**Changes:** L 80.
* * *
**Reviewer:** L74 83% being steeper than 15 degrees is not a particularly interesting stat for a steep mountain basin. Perhaps the authors could add some binning or a steeper threshold?

**Authors:** We have included a additional threshold of 30° to better represent the characteristics for a steep mountain basin.

**Changes:** L 82.